Identification and expression pattern of chemosensory genes in the transcriptome of Propsilocerus akamusi

Yan Chuncai
Sun Xiaoya
Cao Wei
Li Ruoqun
Zhao Cong
Sun Zeyang
Liu Wenbin skylwb@tjnu.edu.cn
Pan Lina skypln@tjnu.edu.cn
Tianjin Key Laboratory of Animal and Plant Resistance, Tianjin Normal University , Tianjin , China
Gillespie Joseph
Electronic publication date: 2020 Jul 21
Publication date: 2020
Volume: 8
Electronic Location ID: e9584
Received 2020 Mar 30; Accepted 2020 Jun 30
Copyright: ©2020 Yan et al.
Copyright year: 2020
Copyright holder: Yan et al.
License: This is an open access article distributed under the terms of the Creative Commons Attribution License, which permits unrestricted use, distribution, reproduction and adaptation in any medium and for any purpose provided that it is properly attributed. For attribution, the original author(s), title, publication source (PeerJ) and either DOI or URL of the article must be cited.
License URL: https://creativecommons.org/licenses/by/4.0/

Keywords: Propsilocerus akamusi (Tokunaga), Transcriptome, Chemosensory genes, Expression pattern

Funding: National Natural Science Foundation of China 31702058 31672324 31801994 Natural Science Foundation of Tianjin 18JCYBJC96300 17JCQNJC14900 14JCQNJC14600 18JCQNJC14700 Tianjin Normal University Foundation 5RL104 043135202-XB1715 52XB1003 52XB1005 043135202-XK1706 135305JF79 This project was supported by the National Natural Science Foundation of China (No. 31702058, 31672324, 31801994), the Natural Science Foundation of Tianjin (No. 18JCYBJC96300, 17JCQNJC14900, 14JCQNJC14600, 18JCQNJC14700) and the Tianjin Normal University Foundation (5RL104, 043135202-XB1715, 52XB1003, 52XB1005, 043135202-XK1706, 135305JF79). The funders had no role in study design, data collection and analysis, decision to publish, or preparation of the manuscript.

==============================
Chironomidae is the most ecologically diverse insects in aquatic and semi-aquatic habitats. Propsilocerus akamusi (Tokunaga) is a dominant and ubiquitous chironomid species in Eastern Asia and its morphologically unique larvae are also considered as indicator organisms to detect water contamination, potential toxicity and waterborne pathogens. Since few studies to date have focused on the olfactory system of P. akamusi, our study aims to elucidate the potential functions of chemosensory genes in P. akamusi. In our study, we found that although signals released from male groups might attract female swarmers, it was a completely male-dominated mating process. Sequencing the transcriptome of P. akamusi on an Illumina HiSeq platform generated 4.42, 4.46 and 4.53 Gb of clean reads for heads, legs, and antennae, respectively. 27,609 unigenes, 20,379 coding sequences (CDSs), and 8,073 simple sequence repeats were finally obtained. The gene-level differential expression analysis demonstrated variants among three different tissues, including 2,019 genes specifically expressed in heads, 1,540 genes in legs, and 2,071 genes in antennae. Additionally, we identified an assortment of putative olfactory genes consisting of 34 odorant binding proteins, 17 odorant receptors, 32 gustatory receptors, 22 ionotropic receptors, six chemosensory proteins as well as 3 sensory neuron membrane proteins; their relative abundances in the above three tissues were also determined by RT-qPCR. Our finding could allow a more plausible understanding of certain olfaction-mediated behaviors in groups of this macroinvertebrate.

Introduction

Propsilocerus akamusi (Tokunaga) is one of the most ubiquitous chironomid species emerging from numerous eutrophic lakes in Eastern Asia. Acting as prey species for fish and aquatic birds as well as decomposers of plants, this kind of midge is able to connect aquatic and terrestrial food webs (Zheng et al., 2017). Considering the relatively high density and richness of P. akamusi in benthic regions, the presence, absence or quantity of their larvae could be a valuable indicator for water quality issues at the organism, population, community, and ecosystem levels (Hirabayashi et al., 2003). The term, Chironomidae, is derived from a Greek word for “pantomimist” due to a typical posture that adult insects tend to have their first pair of legs held forward and upward during the rest time (Meigen, 1803). The up- and outstretched forelegs are thought to assemble antennae and probably act functionally as sensory organs (Armitage, 1995). Males will normally aggregate themselves into a great swarm tending to form above tree-tops, objects and even persons for attracting female ones, which is an extremely common swarm-based mating system in nature (Sæther & Wang, 1996). Plentiful investigations of P. akamusi have been pronounced but most of them primarily focused on its ecology, behavior, karyotype structure, and toxicological response to cadmium stress (Cao et al., 2014; Hirabayashi et al., 2003; Kiknadze, Wang & Istomina, 2009; Zheng et al., 2017). However, limited researches to date have been documented regarding the chemosensory system which actually has a critical role in recognizing chemical cues in the surrounding environment. Since the excessive population of P. akamusi is problematic to other residents living in the shared community, we assume that a better understanding of the chemosensory system could also enable investigators to seek feasible strategies for pest management. The physiological response to certain chemical stimuli is defined as chemoreception and the roles of which in insects are involved in the process of searching food, locating hospitable ovipostion sites, and socializing with other members. A plethora of chemosensory proteins are categorized into receptors and non-receptor proteins, which are believed to account for chemical communication as well as perception (Leal, 2013; Zhao et al., 2016).

Three major groups of the receptor family have been reported, namely odorant receptors (ORs), gustatory receptors (GRs), and ionotropic receptors (IRs). ORs, classified as 7-pass transmembrane proteins, are capable of discriminating distinct odors through its odorant binding sites primarily in sensory neurons. The perception of these chemical stimuli and the consequent intracellular response are associated with multiple behaviors, such as mating, identifying food resources, or alarming conspecies (Venthur & Zhou, 2018). Interface between the surrounding environment and insect taste system could be successfully established by GRs in gustatory neurons since this G protein-coupled receptor is able to detect non-volatile information (Venthur & Zhou, 2018). Chemosensory IRs, a novel family evolved from iGluR-related proteins, are thought to accumulate in sensory dendrites and have unique odor-evoked patterns, which means some of their potential ligands are quite different from those for ORs (Rytz, Croset & Benton, 2013).

In addition to the above-mentioned receptor families, odorant binding proteins (OBPs), chemosensory proteins (CSPs), and sensory neuron membrane proteins (SNMPs) (Jia et al., 2018; Wang et al., 2017a) are also involved in the peripheral olfactory recognition and referred to as non-receptor proteins. OBP family contains a large group of divergent members which are soluble and highly abundant in the sensillum hemolymph, a hydrophilic environment where these non-receptor proteins function as necessary transporters for volatile chemicals and deliver them to activate ORs (Leal, 2013). However, some OBP genes were proved to have significantly higher expression level in the antennae and the rest vary across different tissues (Liu et al., 2020; Zhang et al., 2020). The CSPs own their names due to the fact that these soluble polypeptides are widely expressed in chemosensory tissues of insects, including the antennae, maxillary, labial palps, and proboscis. According the previous reports, some of them have similar functions as OBPs in initiating biochemical recognition while others could perform tasks unrelated to chemosensation, such as development and transition (Liu et al., 2014; Qu et al., 2020; Waris et al., 2020; Zheng, Xia & Keyhani, 2020).

Next generation sequencing technology is now routinely applied for identifying candidate genes since it offers a more efficient and cost-effective way than conventional homology-cloning methods. Therefore, this powerful platform has enabled genes involved in chemoreception to be discovered in some insect species without the availability of entire genomes, including Trichogramma chilonis (Liu et al., 2018), Adelphocoris suturalis (Cui et al., 2017), Bemisia tabaci (Wang et al., 2017b), Periplaneta Americana (Chen et al., 2016), Aphis gossypii (Dickens et al., 2014), Spodoptera littoralis (Gu et al., 2015), and Dendroctonus ponderosae (Andersson et al., 2013).

With reference to previous researches, we hypothesize that P. akamusi behaviors could be more or less manipulated by chemosensory system. Therefore, the genomic information concerning chemoreception and its proper interpretation could offer us valuable insights on this species, which has still been an uncharted territory. Transcriptomic analysis coupled with quantitative real-time PCR were utilized in our study to screen for chemosensory-related genes and their expression patterns, aiming to elucidate the potential functions of these candidate genes.

Materials & Methods

Insect and tissue collection

Males of P. akamusi midges were collected from Tianjin, China, between October 2017 and November 2017 and then transported live to the lab. All samples were transferred within a few hours of eclosion. The antennae, heads (excluding antennae), and legs were excised from 1500 male specimens under microscope for transcriptome sequencing. The obtained tissues were then immersed in RNAlater (Ambion, AM7020) and stored at −20 °C until processing.

Olfactometer bioassay

The emerging P. akamusi males tend to congregate into cloud-like swarms which usually appear in the early evening. They seek to mate when females are around or just pass through, thus starting the life cycle. However, we still face deficient knowledge of the sex differences in mate preferences and wonder whether males will take the first step to attract females or not.

A mate selection behavioral test was therefore conducted using a horizontal Y-tube olfactometer consisting of a central glass tube with the length of 115 mm as well as two lateral glass arms with the length of 75 mm. The diameter for both the tube and arms was 22 mm. The test was performed in a dark room with a small incandescent lamp working as the only light source over the olfactometer. One lateral arm of the olfactometer was selected as the experimental arm and the other one is the control. Either the female subjects or the male ones were firstly placed on a piece of filter paper (10 ×10 mm) and this paper was then carefully inserted into the midpoint of the experimental arm. Another piece of paper without any subjects was inserted into the equivalent point in the control arm. After the olfactometer was connected to an assembled shunting device, the air sampler was switched on to pump airflow for 30s and then a test individual with the opposite sex was placed at the entrance of the central glass tube. If this test one might prefer to enter into the experimental arm and stayed there for at least 1min, the chosen arm could be considered as the one with proper pheromones for the insects. However, the midge was allowed to acclimate and make a preference within 5 min. Otherwise, we assumed that no response was given by the insect to the ones with the opposite sex and the trial was ended then. Besides, to prevent light from interfering with the taxis of P. akamusi, the experimental and control arms were swapped after completing the behavioral assays of three individuals. Fifty insects were tested for one trial with triplicates.

RNA preparation and cDNA library construction

The antennae, heads, and legs of 100 male insects were subjected to RNA preparation using TRIzol reagent (Invitrogen, Carlsbad, CA, USA) according to the manufacturer’s protocol. After the treatment of DNase I, the mRNA molecules were harvested with Oligo (dT) from the total RNA. The long material of mRNA was then shortened into something compatible for further sequencing with the supplementation of Thermomixer buffer. With TransScript First-Strand cDNA Synthesis SuperMix (TransGen, Beijing, China), the fragmented mRNA next worked as the template for single-strand cDNA which was subsequently synthesized to double strand cDNA. Purification, end reparation, adaptor-ligation and size selection were carried out for final PCR amplification. During the QC (quality control) steps, Agilent 2100 Bioanaylzer and ABI StepOnePlus Real-Time PCR System were used for quantification and qualification of the sample library. The library ended up being sequenced using Illumina HiSeq 4000.

De novo transcriptome assembly

After sequencing, a preprocessing of the raw data was performed and consisted of a series of steps, including the trimming of adapter sequences, the removal of reads containing more than 5% ambiguous bases and the elimination of low quality basis. Here the low quality sequences referred to a situation where a single read comprised over 20% of bases with Q score less than 15. The remaining reads generated by these filtering procedures were defined as high quality clean reads and were consequently used for de novo assembly by Trinity v3.0 program (Eisen et al., 1998). This transcript assembly was then clustered into unigenes with the assistance of Tgicl (version: v2.0.6) (De Hoon et al., 2004).

Functional annotation of unigenes as well as prediction of coding sequences and simple sequence repeats

A homology search of functionally annotated unigenes was performed by BLAST (version: v2.2.23, website: http://blast.ncbi.nlm.nih.gov/Blast.cgi) against NCBI non-redundant protein sequences (NR), non-redundant nucleotide sequences (NT), Clusters of Orthologous Groups of Proteins (COG), Kyoto Encyclopedia of Genes and Genomes (KEGG), Gene Ontology (GO) and Swiss-Prot databases (Altschul et al., 1990; Conesa et al., 2005). Besides, InterPro analysis was measured by InterProScan5 (version: v5.11–51.0, website: https://code.google.com/p/interproscan/wiki/Introduction) for providing predictive information of gene families and their critical domains (Quevillon et al., 2005). TransDecoder (https://github.com/TransDecoder/TransDecoder/releases) was utilized to recognize candidate coding regions and peptide sequences within the Trinity-created transcript assembly. Unigenes that could not be aligned to any of the databases mentioned above were later predicted by ESTScan (Saldanha, 2004).

Differentially expressed gene analysis

The clean reads were mapped to unigenes using Bowtie2 (Langmead & Salzberg, 2012), and the expression levels of these high-quality unigene sets were computed based on RSEM values (Li & Dewey, 2011). Poisson distribution was applied to accurately describe gene expression variation on those with more than 2-fold changes as well as less than 0.001 false discovery rate [FDR] (Audic & Claverie, 1997). The performance of hierarchical clustering enabled the differentially expressed genes (DEGs) with similar features to be partitioned into distinct clusters and this analysis was done by having the aid of pheatmap function in R. When two or more relatively homogeneous groups were clustered, the intersection and union DEGs between them were performed. The DEGs were classified according to the GO and KEGG annotation results; FDR was calculated for each p-value. In general, the terms with FDR no more than 0.001 were defined as significantly enriched ones.

Identification of chemosensory genes

To identify candidate chemosensory unigenes for P. akamusi, an analysis using the tBLASTn modules was performed with reference to all the publicly available sequences of OBP, OR, CSP, GR, IR, and SNMP from Diptera species. Meanwhile, all the candidates were manually checked by using the BLASTx program.

RT-qPCR analysis

Using previously obtained cDNA from the three body parts of P. akamusi (the antennae, heads excluding antennae, and legs) as templates, OBPs, ORs, CSPs, GRs, IRs, and SNMPs were selected for RT-qPCR analysis. Specific primer pairs were designed by Primer5 according to the transcriptome data (Table S1). The RT-qPCR was conducted on the Roche LightCycler 480 detector (Stratagene, La Jolla, CA, USA) with the following cycling parameters: 94 °C for 30 s, 40 cycles of 94 °C for 5 s, 55 °C for 10 s, and 72 °C for 10 s. The dissociation curves were carried out as a post-qPCR analysis, during which all the components were firstly denatured for 5s at 95 °C, followed by cooling to 60 °C for 1 min. An increase to 95 °C for 30 s then took place, followed by cooling to 50 °C for 30 s. The relative gene expression data were analyzed by normalizing the threshold cycle (Ct) value of each sample to that of endogenous beta-tubulin, which was determined with 2−ΔΔCt method (Livak & Schmittgen, 2001). The gene amplification from each tissue part of P. akamusi was actually completed in triplicates and the statistical significance was examined with one-way ANOVA test.

Phylogenetic analysis and classification of Chemosensory Genes

A set of chemosensory gene sequences from different Dipteran species were retrieved from the GenBank database, including Drosophila melanogaster, Aedes aegypti, Anopheles gambiae and Culex quinquefasciatus. To visually demonstrate the relationship between candidate chemosensory genes from P. akamusi and other dipteran species, they were first joined with alignments by Muscle 3.8.31 (Edgar, 2004) with default option and then manually refined by BioEdit v7.2.5 (Hall, 1999). The unrooted phylogenetic trees of each family were then generated through running MEGA 7.0 (Kumar, Stecher & Tamura, 2016) and using neighbor joining (NJ) method with Poisson correction, pairwise deletion, and rapid bootstraps (1,000 replicates).

Results

Mate selection behavioral test

To examine whether the male individuals will attempt to mate with female ones, a female insect was placed in the experimental corridor of the Y-tube olfactometer while a male was placed at the open end of the central glass tube. The result showed that the majority of males were relatively active with approximately 58% choosing the female-occupied corridor and 38% choosing the empty side (Fig. 1A). However, this obvious response was not observed when females were given the choice to make a behavioral preference. It turned out that 82% female insects were recorded as non-responders to either corridor and the probability of females selectively orientating to the arm with male subjects was only 8% (Fig. 1B). These results revealed that males tended to seek companions and dominated the mating process probably because of the exposure to airborne pheromones released from the females. We therefore selected male individuals as the main subject for our further investigations and tried to explore how exactly the mutual attraction between the sexes occur during mating.

Figure 1 The movement situation (mean ± SD) of P. akamusi.

(A) The movement situation (mean ± SD) of male P. akamusi when attracted to female. (B) The movement situation (mean ± SD) of female P. akamusi when attracted to male.

Transcriptome sequencing and de novo assembly

With the support of high-throughput sequencing platform, a set of raw sequencing reads were constructed from three different tissues of P. akamusi males, including 50.20 Mb from heads (excluding antennae), 51.82 Mb from antennae and 55.06 Mb from legs (Table 1). The trimming and elimination of adaptor-polluted and low-quality basis led to the generation of 4.42Gb, 4.46Gb, and 4.53 Gb clean reads from heads, legs, and antennae, respectively. The data also showed that the assessed intrinsic quality of the clean reads was sufficient for further analysis (Table 2) and the transcript length distribution was identified as well (Figs. S1, S2). In general, 27,609 unigenes were assembled with a total length, mean length, N50, and GC content of 28,312,956, 1,025 bp, 1,832 bp and 38.23%, respectively (Table 3).

Table 1 Summary of sequencing reads after filtering.

Sample	Total raw reads(Mb)	Total clean reads(Mb)	Total clean bases(Gb)	Clean reads Q20(%)	Clean reads Q30(%)	Clean reads Ratio(%)	
A	50.2	44.17	4.42	98.09	94.38	87.99	
B	55.06	44.55	4.46	97.78	93.6	80.92	
C2	51.82	45.28	4.53	98.12	94.35	87.39	

Table 2 Quality metrics of transcripts.

Sample	Total number	Total length	Mean length	N50	N70	N90	GC(%)	
A	28,143	24,421,248	867	1,576	900	333	36.96	
B	27,460	19,032,332	693	1,231	637	263	38.99	
C2	25,567	22,046,525	862	1,576	957	326	38.23	
Notes.

N50 a weighted median statistic that 50% of the Total length is contained in transcripts great than or equal to this value

GC (%) the percentage of G and C bases in all transcripts

Table 3 Quality metrics of Unigenes.

Sample	Total number	Total length	Mean length	N50	N70	N90	GC(%)	
A	21,857	21,364,850	977	1673	1015	389	37.12	
B	19,975	16,078,029	804	1375	765	313	38.62	
C2	19,355	19,187,584	991	1667	1103	401	38.21	
All-Unigene	27,609	28,312,956	1025	1832	1146	396	38.23	
Notes.

N50 a weighted median statistic that 50% of the Total length is contained in Unigenes great than or equal to this value

GC (%) the percentage of G and C bases in all Unigenes

Functional annotation and classification of the unigenes

The unigenes were annotated in accordance to seven functional databases and the result of this comparison illustrated that 17,311 unigenes (62.70%) were markedly matched with published proteins in NR database. Similarly, the total of annotated unigenes against known information from NT, Swiss-Prot, COG, KEGG, GO and InterPro was 7,936 (28.74%), 14,614 (52.93%), 7,986 (28.93%), 14,341 (51.94%), 7,501 (27.17%) and 14,752(53.43%), respectively (Table 4).

Table 4 Summary of functional annotation result.

Values	Number	Percentage	
Total	27609	100.00%	
NR	17311	62.70%	
NT	7936	28.74%	
Swissprot	14614	52.93%	
KEGG	14341	51.94%	
COG	7986	28.93%	
Interpro	14752	53.43%	
GO	7501	27.17%	
Overall	18544	67.17%	
Notes.

Overall: the number of Unigenes which be annotated with at least one functional database.

The homology searching against NR database revealed that the annotated sequences of P. akamusi were partly matched to sequences of Aedes aegypti (2,769 matching hits, 16.00%), followed by Aedes albopictus (1,870 matching hits, 10.80%), Culex quinquefasciatus (1,741 matching hits, 10.06%) and Anopheles gambiae str. (1,311 matching hits, 7.57%) (Fig. 2).

GO analysis, an enrichment tool, rendered 42,572 unigenes to be grouped into three functional categories, i.e., Biological Process (18,955 unigenes), Cellular Component (15,232 unigenes), and Molecular Function (8,385 unigenes). There were 25 subcategories in Biological Process with ‘cellular process’ accounting for 22.2% (4,213 unigenes) followed by ‘metabolic process’ making up 18% (3,411 unigenes). 19 sub-categories were included in the classification of Cellular Component and among which ‘groups of cell’ (3,145 unigenes, 20.6%) and ‘cell parts’ (3,121 unigenes, 20.5%) were the most abundant GO terms. For Molecular Function, sequences were predominately assigned to ‘catalytic activity’ (3,309 unigenes, 39.5%) and ‘binding’ (3, 267 unigenes, 39%) (Fig. 3A).

Figure 2 The homology searching against the NR database.

Figure 3 Functional annotation for Unigenes of P. akamusi.

(A) Functional distribution of GO annotation. The x-axis represents the number of Unigenes. The y-axis represents the Gene Ontology functional category. (B)Functional distribution of KEGG annotation. The x-axis represents the number of Unigenes. The y-axis represents the KEGG functional category. (C)Functional distribution of COG annotation. The x-axis represents the number of Unigenes. The y-axis represents the COG functional category. (D)Venn diagram between NR, COG, KEGG, Swissprot and Interpro.

A more detailed comprehension of gene biochemical function could be further gained via KEGG analysis. The result delineated that a total of 25,377 unigenes were mapped and organized to six functional clusters, including Cellular Process (2,476 unigenes), Environmental Information Processing (2,613 unigenes), Genetic Information Processing (2,531 unigenes), Human Disease (6,293 unigenes), Metabolism (6,405 unigenes), and Organismal System (5,059 unigenes) (Fig. 3B).

Besides, potential functions of these putative unigenes were predicted with the assistance of COG database which was considered as a useful tool for understanding the orthologous relationships of gene products. The retrieved unigene sets were classified under 25 categories, among which the cluster of ‘General function prediction only’ (2,338 unigenes, 19%), ‘Translation, ribosomal structure and biogenesis’ (979 unigenes, 8%) and ‘Posttranslational modification, protein turnover, chaperones’ (941 unigenes, 7.7%) were the most representative three classifications (Fig. 3C). Venn diagram aimed at describing the similarities and differences of unigenes when searched against NR, COG, KEGG, Swissprot and Interpro databases and it turned out that 6,766 unigenes overlapped as the intersection proportion (Fig. 3D).

CDS prediction

A total of 17,522 CDSs were screened out from the 27,609 annotated unigenes. Since ESTScan program was capable of constructing CDSs in the remaining unigenes which were not of particular matches with the aforementioned databases, a total of 2,857 CDSs were predicted from these unannotated unigenes. Altogether, 20,379 CDSs were predicted from unigenes with a total length, a mean length, N50 and GC content of 16,667,634 bp, 817 bp, 1,290 bp and 42%, respectively (Fig. 4, Table 5).

Differentially expressed genes

Differentially expressed genes were examined by comparison among heads, legs, and antennae of male P. akamusi midges (Fig. 5A). The gene expression profiling analysis demonstrated that there were 2,281 up-regulated along with 4,865 down-regulated genes in heads against legs. A total of 3,421 were expressed at a markedly higher level while 5,792 genes were at lower levels in antennae vs. heads. Plus, 2,066 genes were expressed with greater numbers while 7,093 genes with relatively smaller numbers in antennae vs. legs. Shared and exclusively expressed genes among three tissues were shown in Fig. 5B. The number of genes differentially expressed among three tissue types was quite similar to each other with 21,103 in heads, 20,980 in legs, and 21,750 in antennae. Meanwhile, a subset of 5,620 genes that differentially expressed were identified with a tissue-dependent manner, including 2019 specifically expressed in head, 1540 genes expressed in legs and 2071 expressed in antennae.

Figure 4 CDS length distribution of P. akamusi.

The x-axis represents the length of CDS. The y-axis represents the number of CDS.

Candidate odorant binding proteins

Based on functional annotation and tBLASTn results with an E-value of 1E-5 or lower (Table S2), 34 transcripts ranging from 138 to 927 bp were isolated as best candidate OBPs (PakaOBPs) in the P. akamusi transcriptome, and 29 of which contained full-length open reading frames (ORFs). The identified PakaOBP transcripts together with corresponding sequence data of OBPs from A. gambiae, C. quinquefasciatus, D. melanogaster and A. aegypti were used for obtaining phylogenetic inferences, as depicted in Fig. S3.

To determine the transcriptional output of these candidate genes among different tissues, 12 PakaOBPs with RPKM>1.2 were selected for RT-qPCR amplification and the expression profile of each gene differed. Five OBPs (PakaOBP1, 5, 8, 9 and 10) had noticeably higher proportion in antennae than either heads or legs. Meanwhile, seven other OBPs (PakaOBP1, 2, 3, 4, 6, 7, and 12) were expressed with a markedly greater numbers in legs. Of note, we found a unique antennae-specific expression pattern of PakaOBP9 since this particular gene showed almost no evidence of presence in other tissues (Fig. 6).

Table 5 Quality metrics of predicted CDS.

Software	Total number	Total length	Mean length	N50	N70	N90	GC(%)	
Blast	17,522	15,693,411	895	1,350	912	417	41.83	
ESTScan	2,857	974,223	340	339	261	213	44.69	
Overall	20,379	16,667,634	817	1,290	834	348	42.00	
Notes.

N50 a weighted median statistic that 50% of the Total length is contained in CDS great than or equal to this value

GC (%) the percentage of G and C bases in all CDS

Figure 5 Differentially expressed genes (DEGs) in P. akamusi.

(A) Summary of DEGs. The x-axis represents comparing samples. The y-axis represents the number of DEGs. Red color represents up-regulated DEGs while blue color represents down-regulated ones. Group: A, Heads; B, Leg; C, Antennae. (B) Venn diagram showing intersections and disjunctions of genes expressed in heads, legs and antennae.

Figure 6 P. akamusi OBPs transcript levels in different tissues measured by RT-qPCR.

“AN” for antennae; “HE” for head; “LE” for leg. The tubulin was used to normalize transcript levels in each sample. The standard error is represented by the error bar, and the different letters (a, b, c, d) above each bar denote significant differences (p < 0.05).

Candidate odorant receptors

Putative OR genes of P. akamusi (PakaORs) were represented based on their similarities to known insect ORs and tBLASTn results with an E-value of 1E-5 or lower generated a total of 17 PakaORs with length ranging from 312 to 1,671 bp (Table S3). Among them, 16 sequences were available as full-length coding ones. Evolutionary distances were evaluated among ORs from the sampled P. akamusi and four other Dipteran species, as shown in Fig. S4.

RT-qPCR was performed for 6 PakaORs that were relatively abundant in the antennal transcripts with RPKM>1.2. All of these PakaOR genes except for PakaOR4 were highly expressed in antennae compared with heads and legs whereas PakaOR3–6 had a relatively sufficient expression only in heads (Fig. 7). The results of PakaOR expression appeared to be partially consistent with those from the RPKM analysis.

Figure 7 P. akamusi ORs transcript levels in different tissues measured by RT-qPCR.

“AN” for antennae; “HE” for head; “LE” for leg. The tubulin was used to normalize transcript levels in each sample. The standard error is represented by the error bar, and the different letters (a, b, c, d) above each bar denote significant differences (p < 0.05).

Candidate gustatory receptors

A total of 32 candidate GR genes (PakaGRs) were identified based on the tBLASTn results with an E-value of 1E-5 or lower, with 20 sequences containing a full-length ORF (Table S4). A phylogenetic tree was constructed by using concatenated orthologous sequences derived from GRs in different Dipteran species (Fig. S5).

The relatively abundant transcripts (PakaGR1–10) according to the RPKM analysis were subjected to RT-qPCR detection, aiming to figure out the particular tissues where they preferred to be expressed. It turned out that PakaGR1 and PakaGR2 were mainly expressed in the antennae whereas PakaGR3 had a significant quantity in legs. Meanwhile, a pervasive expression pattern was shown for 10 other PakaGRs, including PakaGR2 and PakaGR3–10 (Fig. 8).

Figure 8 P. akamusi GRs transcript levels in different tissues measured by RT-qPCR.

“AN” for antennae; “HE” for head; “LE” for leg. The tubulin was used to normalize transcript levels in each sample. The standard error is represented by the error bar, and the different letters (a, b, c, d) above each bar denote significant differences (p < 0.05).

Candidates for other chemosensory genes

Our bioinformatic data facilitated the discovery of 22 transcripts for candidate IR genes (PakaIRs), 6 for candidate CSP genes (PakaCSPs), and 3 for candidate SNMP genes (PakaSNMPs) with the result of tBLASTn being E-value of 1E-5 or lower; of these genes, 19 PakaIRs, 2 PakaCSPs, and 3 PakaSNMPs included full length ORFs (Tables S5, S6). The phylogenetic signals of PakaIRs, PakaCSPs, and PakaSNMPs from P. akamusi with those from other four species were mapped onto the tree, as described in Fig. S8.

RT-qPCR data revealed that 16 PakaIRs were highly expressed in heads and legs (Fig. 9) while CSP1 was distributively expressed at varying levels in all tissues. Additionally, there was an abundance of PakaSNMP1 presented in antennae and legs at transcript level, which was a reliable clue for understanding its role in chemosensory process.

Figure 9 P. akamusi IRs transcript levels in different tissues measured by RT-qPCR.

“AN” for antennae; “HE” for head; “LE” for leg. The tubulin was used to normalize transcript levels in each sample. The standard error is represented by the error bar, and the different letters (a, b, c, d) above each bar denote significant differences (p < 0.05).

Discussion

The Chironomidae chemoreception is barely explored at either genetic or molecular levels, especially in comparison to some other Dipterans, such as Culicidae and Drosophila, which are commonly selected as subjects in researches of chemosensation.

Next generation sequencing technology was utilized in our study to yield a diverse array of candidate chemoreception-related genes of P. akamusi, a dominant Chironomidae species in freshwater environment. The sequencing data presented six categories of putative chemosensory genes, including 34 PakaOBPs, 17 PakaORs, 32 PakaGRs, 22 PakaIRs, 6 PakaCSPs, and 3 PakaSNMPs (Figs. 6–10), which was the first known attempt for the identification of chemosensory genes in P. akamusi. However, it turned out that the number of these identified genes was quite tiny when compared to that of genes from other species. For instance, 82 ORs, 77OBPs, 30GRs, and 102IRs have previously been collected from Aedes albopictus. Strictly speaking, we have to admit that most of our transcripts did not comprise complete ORFs.

Figure 10 P. akamusi CSP1 and SNMP1 transcript levels in different tissues measured by RT-qPCR..

“AN” for antennae; “HE” for head; “LE” for leg. The tubulin was used to normalize transcript levels in each sample. The standard error is represented by the error bar, and the different letters (a, b, c, d) above each bar denote significant differences(p < 0.05).

The genome sequencing in our work facilitated an identification of 34 PakaOBPs, which was fewer than the numbers reported from other species, including Bombyxmori (44) (Gong et al., 2009), A. gambiae (57) (Biessmann et al., 2002; Vogt, 2002; Xu, Zwiebel & Smith, 2003), D. melanogaster (51) (Hekmat-Scafe, 2002), and Agrotis ipsilon (33) (Gu et al., 2013). We assume that the less quantitative sufficiency of OBPs in various tissues of P. akamusi accompanied with barely detectable levels could be one possible explanation. Besides, the lack of some host-seeking cues in lab could be another factor for this limited number since OBPs will be normally stimulated with the exposure to proper odors in the nature. The tissue-specific patterns of candidate OBPs could provide essential clues for gene function. For example, we observed that PakaOBP9 was specifically expressed in the antennae whereas most PakaOBPs repertoire did not apparently display an antenna-biased expression profile (Fig. 6), suggesting a likely association with certain chemosensory reactions. Besides, PakaOBP1 was abundantly expressed in antennae whereas PakaOBP5 was highly expressed in heads and antennae (Fig. 6). One previous study has reported that OBP21 was highly expressed in a non-olfactory system, like the venom gland of Apis mellifera, and hypothesized that this molecule might serve as a carrier of potential ligands other than odorants (Li et al., 2013; Pelosi et al., 2018). However, the expression profile of PakaOBPs in a non-olfactory part still remains vague and requires a further exploration to better appreciate the function of this gene.

The ORs, belonging to the superfamily of G-protein coupled receptors, are critical recognition elements involved in olfactory sensory system. We have analyzed the whole genome to prospect candidate ORs and finally recognized as many as 11 transcripts in P. akamusi, which is actually far fewer than 170 genes found in Apis mellifera (Robertson & Wanner, 2006) and 60 in the parasitoids Microplitis mediator (Wang et al., 2015) but greater than 6 discovered in Cotesia vestalis (Matsunami et al., 2012b). Most of newly PakaORs transcripts showed a significant enrichment in the antennae whereas PakaOR3–6 were predominantly detected in tissues like heads. The expression pattern of ORs in non-olfactory tissues suggests they might have some undefined physiological functions.

GRs, regarded as critical chemoreceptors, usually endow gustatory receptor neurons with the capability to perceive soluble tastes and respond to carbon dioxide. This kind of peripheral receptor comprises more conserved sequences and structures among diverse species when compared to ORs. The de novo assembly in our work allowed a total of 32 PakaGRs to be captured, the same number as identified in A. Albopictus. Despite the typical antenna-specific expression pattern of GRs in most insect species, our discovery revealed that most PakaGRs were significantly enriched in legs (PakaGR2, 3, 5, 6, 7, 9, and 10) or heads (PakaGR5 and 9). Since P. akamusi posturing with forelegs is a common phenomenon in nature, we wonder whether the high abundance of ORs in legs might have potential roles in chemosensory-related perception.

IRs are novel families of highly divergent ionotropic glutamate receptors and broader attentions with respect to this group have been constantly received. 66 have been reported in D. melanogaster (Benton et al., 2009), 21 in Manduca sexta (Koenig et al., 2015), 15 in Cydia pomonella (Matsunami et al., 2012a), and 12 in Helicoverpa armigera (Dickens et al., 2015). Although IRs have been reported to exist in various insect genomes or transcriptomes, their function has only been hinted in studies focusing on Drosophila. There is a considerable discrepancy between the environmental stimuli recognized by ORs and IRs. In a combinatorial fashion, IRs could provide a strong response to a broad range of odors which, however, only induce weak or even no answer of ORs. We have identified a set of 22 IRs in the genome of P. akamusi compared to transcriptomes of D. melanogaster with 66 IRs (Benton et al., 2009), Manduca sexta with 21 IRs (Koenig et al., 2015), Cydia pomonella with 15 IRs (Matsunami et al., 2012a; Matsunami et al., 2012b), and Helicoverpa armigera with 12 IRs (Dickens et al., 2015). A comprehensive map of their expressions was generated with reference to RPKM values and we found that PakaIR3, 4, 7, 8, 9, 10, 12, and 13 showed high transcript levels in heads and legs while none in antennae. Plus, the expression levels of these PakaIRs are higher than those of PakaOBPs and PakaCSPs, indicating possible protein functions in chemosensory processes in these non-olfactory parts.

Conclusions

In our study, the transcriptome of P. akamusi was sequenced by using Illumina Hiseq platform by which 27,609 unigenes, 20,379 CDSs, and 8,073 simple sequence repeats were obtained. The differentially expressed gene analysis showed that there were 2,019 head-specific genes, 1,540 leg-specific genes, and 2,071 antennae-specific genes. Furthermore, candidate olfactory-related genes were identified and their relative abundances in the above tissues were examined by RT-qPCR as well. In general, 34 odorant binding proteins, 17 odorant receptors, 32 gustatory receptors, 22 ionotropic receptors, six chemosensory proteins as well as three sensory neuron membrane proteins were collected.

Supplemental Information

Supplemental Information 1 Raw data: ll unigenes of P. akamusi used in this study

Click here for additional data file.

Supplemental Information 2 Raw data of Fig. 1

Click here for additional data file.

Supplemental Information 3 Primers used for RT-qPCR analysis of olfactory genes of the P. akamusi

Click here for additional data file.

Supplemental Information 4 Raw data of Figs. 6–10 Real-time data.xlsx

Click here for additional data file.

Supplemental Information 5 The list and the nucleotide sequences of 34 OBPs of P. akamusi identified in present study

Click here for additional data file.

Supplemental Information 6 The list and the nucleotide sequences of 17 ORs of P. akamusi identified in present study

Click here for additional data file.

Supplemental Information 7 The list and the nucleotide sequences of 32 GRs of P. akamusi identified in present study

Click here for additional data file.

Supplemental Information 8 The list and the nucleotide sequences of 22 IRs of P. akamusi identified in present study

Click here for additional data file.

Supplemental Information 9 The list and the nucleotide sequences of 6 CSPs and 3 SNMP genes of P. akamusi identified in present study

Click here for additional data file.

Supplemental Information 10 Transcript length distribution of P. akamusi

The x-axis represents the length of transcripts. The y-axis represents the number of transcripts.

Click here for additional data file.

Supplemental Information 11 Unigene length distribution of P. akamusi

The x-axis represents the length of Unigenes. The y-axis represents the number of Unigenes.

Click here for additional data file.

Supplemental Information 12 Neighbor-joining tree of OBP amino acid sequences from D. melanogaster

A. aegypti, A. gambiae, C. quinquefasciatus and P. akamusi. Bootstrap values were calculated with 1,000 replicates and the values < 50% are shown on the branches. Yellow: D. melanogaster; Blue: A. gambiae; Purple: C. quinquefasciatus; Green: A. aegypti; Red: P. akamusi.

Click here for additional data file.

Supplemental Information 13 Neighbor-joining tree of OR amino acid sequences in D. melanogaster

A. aegypti, A. gambiae, C. quinquefasciatus and P. akamusi. Bootstrap values were calculated with 1,000 replicates and the values < 50% are shown on the branches. Yellow: D. melanogaster; Blue: A. gambiae; Purple: C. quinquefasciatus; Green: A. aegypti; Red: P. akamusi.

Click here for additional data file.

Supplemental Information 14 Neighbor-joining tree of GR amino acid sequences in D. melanogaster

A. aegypti, A. gambiae, C. quinquefasciatus and P. akamusi. Bootstrap values were calculated with 1000 replicates and the values < 50% are shown on the branches. Yellow: D. melanogaster; Blue: A. gambiae; Purple: C. quinquefasciatus; Green: A. aegypti; Red: P. akamusi.

Click here for additional data file.

Supplemental Information 15 Neighbor-joining tree of IR amino acid sequences in D. melanogaster

A. aegypti, A. gambiae, and P. akamusi. Bootstrap values were calculated with 1,000 replicates and the values ¿50% are shown on the branches. Yellow: D. melanogaster; Blue: A. gambiae; Green: A. aegypti; Red: P. akamusi.

Click here for additional data file.

Supplemental Information 16 Neighbor-joining tree of CSP amino acid sequences in D. melanogaster

A. gambiae, C. quinquefasciatus and P. akamusi. Bootstrap values were calculated with 1,000 replicates and the values >50% are shown on the branches. Yellow: D. melanogaster; Blue: A. gambiae; Purple: C. quinquefasciatus; Red: P. akamusi.

Click here for additional data file.

Supplemental Information 17 Neighbor-joining tree of SNMP amino acid sequences in D. melanogaster

A. aegypti, A. gambiae, C. quinquefasciatus and P. akamusi. Bootstrap values were calculated with 1000 replicates and the values < 50% are shown on the branches. Yellow: D. melanogaster; Blue: A. gambiae; Purple: C. quinquefasciatus; Green: A. aegypti; Red: P. akamusi.

Click here for additional data file.

Additional Information and Declarations

Competing Interests

Author Contributions

Data Availability

The authors declare there are no competing interests.

Chuncai Yan and Lina Pan conceived and designed the experiments, prepared figures and/or tables, authored or reviewed drafts of the paper, and approved the final draft.

Xiaoya Sun performed the experiments, prepared figures and/or tables, authored or reviewed drafts of the paper, and approved the final draft.

Wei Cao and Ruoqun Li performed the experiments, prepared figures and/or tables, and approved the final draft.

Cong Zhao performed the experiments, authored or reviewed drafts of the paper, and approved the final draft.

Zeyang Sun analyzed the data, authored or reviewed drafts of the paper, and approved the final draft.

Wenbin Liu conceived and designed the experiments, analyzed the data, authored or reviewed drafts of the paper, and approved the final draft.

The following information was supplied regarding data availability:

All unigenes of P. akamusi and the list and accession numbers of candidate olfactory-related genes of P. akamusi are available in the Supplemental Files.

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
