# Peer review of "Identification and expression pattern of chemosensory genes in the transcriptome of Propsilocerus akamusi"

_PeerJ, doi:10.7717/peerj.9584_

## Round 0.1 · original submission · Major Revisions

Dear Dr. Yan and colleagues:

Thanks for submitting your manuscript to PeerJ. I have now received three independent reviews of your work, and as you will see, the reviewers raised some concerns about the research. Despite this, these reviewers are optimistic about your work and the potential impact it will have on research studying Propsilocerus akamusi chemosensory genes. Thus, I encourage you to revise your manuscript, accordingly, taking into account all of the concerns raised by all three reviewers.

Importantly, please ensure that an English expert has edited your revised manuscript for content and clarity. Please also ensure that your figures and tables contain all of the information that is necessary to support your findings and observations. Please also make sure that all taxon names are spelled correctly, italicized properly, etc.

Phylogeny estimations should be made with the proper outgroups. The also appears to be important information lacking regarding the PCR assays, particularly tissue information.

Please ensure that all materials and information are thoroughly described such that all analyses may be repeated independently.

I look forward to seeing your revision, and thanks again for submitting your work to PeerJ.

Good luck with your revision,

-joe

Reviewer 1 ·

Basic reporting

The Background part need to be improved. The latest progress of all olfactory genes should be added.

Experimental design

The tissues used to qPCR analysis is quite few.

Validity of the findings

Obtain some novel olfactory genes in the Propsilocerus akamusi. However, the number is quite low.

Additional comments

Major concerns,
1. The phylogenetic trees should be constructed, adding some flies and mosquitos, and then the newly identified gene functions can be inferred.
2. The olfactory gene numbers are remarkably lower than other mosquitos. Why?
Minors,
1. Line 31, Give the full name of CDS
2. Line 37, Change “feet” to “leg.” Revise the whole MS.
3. Line 46, The blank should be added between the reference and the text. Revise the whole MS.
4. Lines 67-82, should add some lastest progress in these genes.
5. Lines 102-104, Did all tissues collect from male adults? And which days after eclosion? Should clarify.
6. Line 186, which dipteran species did you select? Should state the Blast parameters.
7. Line 199, why did you select tubulin as the reference gene? Is it stable across tissues?
8. Line 200, which one-way ANOVA test did you use?
9. Line 280, according to the nomenclature rule, the chemosensory genes in insects should be named PakaOBPs, PakaORs, etc.
10. Line 286, not “interestingly,” many OBPs are expressed only in antennae.

Reviewer 2 ·

Basic reporting

The author carried out a transcriptome sequencing of P. akamusi. This is interesting study because of the ecology and environmental importance of this insect species. These are clearly introduced. After went through the MS, I believe the MS failed to meet PeerJ standards and decided not to check the literature reference (see below).
English needs to be improved throughout the MS to ensure that an international can clearly understand your text. Some examples are given below.
Figures are not very clear. The legend of Figure 2 is completely wrong, which is completely unacceptable because it demonstrates the authors do not read their MS carefully and professionally before submitting to PeerJ.

Experimental design

The experimental design is interesting. The authors started with the mating behavioural study and demonstrated an interesting result; males rather than females exhibit the attraction to females which are insensitive to males. This is another strengthen of this MS. The authors decided to sequencing the transcriptome of male olfactory tissues. Unfortunately, the authors missed out the importance of control experiments. First, the analysis of female tissues should be included along with the male tissues. Second, a non-olfactory tissue should be included for the analysis of gene expression with 2-ΔΔCt method correctly. These make the authors the biological aspect of the data, and the possible correlations between the behavioural results and the transcriptome analysis.
The relative expression levels are presented in all RT-qPCR results as ratio of target gene expression /reference gene expression. This is not correct. In the 2-ΔΔCt method, a reference (1st delta) to normalise the expression of a target gene and non-olfactory tissue (2nd delta) to calculate relative expression as fold change. As the main experimental technique of the current study is the real-time qPCR, I would strongly urge the authors to reanalyse the data by using two reference genes and to study the commonly accepted MIQE guidelines on minimum information for the publication of quantitative real-time PCR experimental data (Bustin et al. (2009) Clin Chem 55,611).

Validity of the findings

The presence of chemiosensory genes in the genomes of Dipterans is not surprising. The identification of them in P. akamusi is expected and not novel. The knowledge gap that this study provides in more such genes and their expression in olfactory tissues is makes the MS becomes another descriptive transcriptome sequencing MS, and provides no advance in olfaction research.

Additional comments

The description of Methods such as cDNA synthesis, bioinformatics analyses is too generalized. More details related to this insect species and experimental conditions are needed. Some examples are given below.
For above reasons, I would not recommend for the MS to be accepted for the publication in PeerJ.

Other points are listed below.
Line 32. “gene analysis showed little difference …in the three types of tissues”. However, DEG analysis shows > 1500 differential expression. This is not little difference.
Line 37. “17 odorant receptor (head 4; antennae 5)…”. The number in () does not match the total number.
Line 47. What dose “their high species richness” means?
Line 103. I suggest include another non-olfactory tissue for RT-qPCR analysis correctly.
Line 108. Change “fly” to “flies”. Please check all grammar mistakes in the MS.
Line 116. Please check grammar mistake.
Line 116. Experimental design. How to keep a live insect in an arm to attract another insect.
Line 134. “cDNA was synthesized …”. Please give more detail.
Line 135. “short fragments were purified…”. Please give details how they are purified and under what condition.
Line 138. Please give more details on quantification and qualification of the libraries.
Line 147. what does "low quality reads, defined as those in which >20% of the base had a quality <15” mean?
Line 150-154. Please do not explain how Trinity works if you have no idea how it exactly works.
Line 155-156. So the outcome of your Trinity assembly is a list of unigenes with either CL or unigene as their prefix. Is it true?
Line 161. How did you use BLAST to annotate unigenes into COG and GO databases? Please all websites of the software you used.
Line 165-166. This not the way to define CDS of a gene.
Line 168. what is the rational for this analysis? Do SSRs involved in insect olfaction?
Line 198. a reference gene (1st delta) and non-olfactory tissue (2nd delta) should be used. In this method, a reference (1st delta) to normalise the expression of a target gene and non-olfactory tissue (2nd delta) to calculate relative expression as fold change.
Line 199. Two reference genes should be used.
Line 205. Please check English in “To test the mate selection behavior reaction of individual male insects”
Line 232. What do you mean by “the distribution of annotated species is statisticed”
Line 262-263. An interesting result on SSRS but what do they implicate? any olfactory function.
Line 268. The feet are also sensory organ as you stated in the Introduction so it is better to compare with a non-olfactory tissue.
Line 279-280. What criteria to define “34 transcripts that encoded candidate OBPs”
Line 248. “with five expressed higher in the antennae” relative to which tissue? Please check other sentences.
Line 298. Please change “transcriptomes” to transcripts.
Line 311. I would be more interested in the comparison between these three SMNPs with those of mosquitoes and Drosophila.
Line 335. incomplete discussion, which PaOBP is homologous to AmelOBP21?
Line 387-388. All transcriptoe sequences and unigene sequences should be deposited into a GenBank database.
Table 1, Table 2 and Table 3. What do “A, B, C2” stand for?
Figure legends need more detail description.
Figure 2 has a wrong legend.
Figure 6. It is not clear what comparison the Figure displays. The DEGs are relative.
Figure 7-11 (all RT-qPCR results). Relative expression of a gene should be fold change between target tissue and reference tissue after the normalization with the expression of the reference gene.

Reviewer 3 ·

Basic reporting

Please see the "General comments for the author"

Experimental design

Please see the "General comments for the author"

Validity of the findings

Please see the "General comments for the author"

Additional comments

The manuscript by Yan and colleagues surveys a transcriptome assembly generated with Illumina HiSeq platform of P. akamusi, which obtained 27,609 unigenes, 20,379 CDSs. In addition, they also identified some candidate olfactory-related genes and their relative abundances in the three tissues (antennae, head and feet). The organism is important and the goals of the study are sound. Although I feel that there is some value to this study, there are some methodological and analytical shortcomings that the authors must address before a decision can be made about the suitability of this manuscript for publication. My comments are below.

1. The abstract does not tell the reader anything about mate selection behavioral test which is the first part of your results. It should be concerned in the abstract.
2. Some titles in the “Materials & Methods” section are the same as the titles of the “results”, such as “2.2. Mate selection behavior test” and “3.1. Mate selection behavioral test”.
3. The authors must state here how many individuals were subjected to the RNA preparation and cDNA library construction.
4. The integrity of the RNA samples (e.g. RIN) is not given. This becomes crucial in the results as there are concerns about RNA integrity.
5. What is “clean reads” (line 222) ? Please clarify.
6. No software version numbers or parameters are given for the de novo assembly the downstream clustering with tgicl. More details on the specific run-time parameters used should be given in all instances.
7. It is recommended to unify the expression of “leg”, in some sentence the author used “foot”.
8. Please recheck the comments of Fig.2 and Fig.3.

---

## Round 0.2 · accepted · Accept

Dear Dr. Yan and colleagues:

Thanks for revising your manuscript based on the concerns raised by the reviewers. I now believe that your manuscript is suitable for publication. Congratulations! I look forward to seeing this work in print, and I anticipate it being an important resource for groups studying Propsilocerus akamusi chemosensory genes. Thanks again for choosing PeerJ to publish such important work.

Best,

-joe

Reviewer 3 ·

Basic reporting

The authors have made a lot of changes and improvements. It is suggested to accept.

Experimental design

no problem.

Validity of the findings

The authors have made a lot of changes and improvements. It is suggested to accept.

Additional comments

The authors have made a lot of changes and improvements. It is suggested to accept.